# The Influence of Online Reviews on the Purchasing Decisions of Travel Consumers

**Qin-Min Wu**

Department of Management Science, Fudan University, Shanghai 200433, China; 17110690013@fudan.edu.cn or qmwuu@fudan.edu.cn

**Abstract:** In this study, we investigate the impact of online review characteristics on consumers' purchasing decisions in the context of spatial distance. We consider the product experience of online travel routes, geographical location characteristics, and price adjustment factors, as well as the dynamics between consumers and businesses during the booking of travel routes. Through empirical research and large-scale data simulation experiments, we have found that the variability in attributes of tourist routes significantly influences the user recommendation rate, while the overall rating has a positive moderating effect. Furthermore, the number of reviews negatively moderates the relationship between them. Additionally, the product information and service quality of tourist routes also significantly affect the recommendation rate. Finally, we propose a management strategy for tourism route managers to enhance user recommendation rates and achieve greater benefits.

**Keywords:** online reviews; purchase decisions; travel consumer; game strategies

## 1. Introduction

With the rapid advancement of the Internet, big data, and artificial intelligence, Internet applications continue to evolve. The online hotel booking industry is also experiencing innovation. In comparison to traditional media, the Internet possesses stronger communication capabilities, broader coverage, and a wider audience. Online reviews disseminated through Internet platforms offer distinct advantages and influence that traditional word-of-mouth does not provide. Consequently, this aids consumers in making informed judgments and enhances the efficiency of their decision-making.

Online reviews play a pivotal role in the decision-making process of consumers when booking hotels online. This paper specifically addresses this issue and examines how these reviews impact the purchasing decisions of hotel consumers [1].

We examine the impact of psychological distance on hotel consumers' purchase intention by integrating online hotel product experience, geographical location characteristics, and price adjustment. We explore the purchase intention process and influencing factors of hotel consumers before purchase and analyze the influence of different types of online reviews on the purchase intention of hotel consumers under varying psychological distance and price adjustment scenarios. The findings indicate that rational online reviews have a more significant impact on consumers' purchase intention when they are in the most recent consumption period or when social or spatial distance is relatively close. Conversely, emotional online comments have a more pronounced impact in different situations. Moreover, when hotels implement price reduction promotions, rational online reviews have a more significant impact on the purchase intention of psychologically close consumers.

Based on the dynamics of the game process between consumers and hotel enterprises in the hotel booking process, we construct a two-stage game model to analyze the impact of online review characteristics on the purchase decision of hotel consumers in the context of spatial distance [2]. We propose corresponding game strategies and analyze the effectiveness of hotel consumers in hotel product booking. Utilizing obtained hotel

big data, simulation experiments are conducted to analyze the impact of the overall hotel rating and the number of reviews on consumer utility when hotel prices change. The simulation results demonstrate that both the overall rating and the number of reviews have a positive impact on consumer utility, with the overall rating exerting a higher influence than the number of reviews. While the number of reviews also positively affects purchasing decisions, its effect is relatively less pronounced compared to overall ratings. Building on this research, we present several strategies for the presentation of hotel online reviews and guiding consumers to score, aiming to assist hotels or online booking platforms in improving efficiency and contributing to the development of hotel e-commerce.

Online reviews play a significant role in influencing consumers' shopping decisions. This article can provide valuable insights for both tourism consumption and hotel enterprises to enhance their benefits. By conducting mechanism analysis and empirical research, this paper explores the impact of online reviews on consumers' purchasing decisions. It marks the first empirical research on consumer decision rules in the field of e-commerce, signifying a crucial expansion in this area. The findings of this study offer new ideas and research methods for the field of consumer behavior and hold reference significance for enterprises. While both consumers and platform managers have found meaningful insights, there are still areas that require further research and improvement.

The research outcomes presented in this paper are sustainable, and future studies could consider the influence of multidimensional psychological distance on hotel consumers' purchase intentions. By incorporating the review content and scores of various hotel attributes, the model can be enhanced to yield more accurate results, facilitating a deeper understanding of the impact of online reviews on consumer purchase decisions. Considering the diverse styles and characteristics of hotels, the influence of different types of consumers on their post-purchase recommendation decisions, including the number and content of review texts, can be analyzed. Additionally, the impact of hotel property differences on the purchase intentions of various consumer types can be explored.

The remaining sections of the paper are organized as follows: Section 2 presents a literature review, Section 3 outlines the research methods, Section 4 presents the game between consumers and travel routes, Section 5 discusses game strategy and equilibrium, and Section 6 covers simulation and result analysis. Finally, Section 7 summarizes the conclusions.

## 2. Literature Review

### 2.1. Consumer Utility and Two-Stage Dynamic Pricing Model

According to consumer utility theory, it is more reasonable for manufacturers to analyze consumer utility in order to determine product demand when making production or pricing decisions. For instance, Lancaster [3] introduced the product vertical difference model, which established product demand when consumers had consistent preferences for products from the perspective of consumer utility. The author used optimization theory to analyze the optimal pricing decision of oligopoly. Subsequently, many researchers utilized the consumer utility model to investigate various decision-making problems of manufacturers, such as Atasu [4], Choudhary [5], Rao [6], and Pasquale [7]. When making purchase decisions, consumers assess potential purchase schemes based on utility. In subsequent decision-making behaviors, consumers are more inclined to transact with merchants who can offer them the greatest value. Chen et al. [8] and Kim et al. [9] empirically verified that consumers' subjective perception of overall utility determines their purchasing decisions.

As a unique form of dynamic pricing model, the two-stage dynamic pricing model is frequently employed to illustrate specific effects or consumer behaviors associated with temporal dynamics. This paper adopts the definition of consumer strategic behavior used by Cachon et al. [10], Han [11], Kumar [12], Mudambi [13], and Park [14]. Strategic consumers, based on future inventory and price expectations, compare the anticipated gains from immediate purchases with those from delayed purchases in order to select the timing that maximizes their profits. The sales period is divided into two phases by the

strategic consumer. It is assumed that the price of the product in the initial phase of the interaction between the retailer and the consumer is represented as p, while the price in the subsequent phase is denoted as s (assuming that the price change trend adheres to the pricing pattern of seasonal products), and the consumer's perception and evaluation of the product is represented as $v$. The consumer surplus in the first phase of the interaction is $v - p$. The strategic consumer anticipates that the product price will drop to s and, therefore, compares the consumer surplus, $v - p$, from an immediate purchase with the consumer surplus, $\varphi(\delta v - p)$, from waiting for the next purchase, where $\varphi$ represents the rational expectation of the probability that the consumer will obtain the product in the next purchase, and $\delta$ denotes the discount factor for the consumer's product value evaluation. When a product is in stock with a consumer surplus of $v - p \geq \varphi(\delta v - s)$ during the current period, the strategic consumer will opt to make the purchase at that time [15].

In the process of consumers purchasing tourism products, there is a certain game between consumers and tourism products due to information asymmetry. On the one hand, travel companies hope to maximize their profits through their pricing; consumers, on the other hand, take price, location, reviews, and other factors into consideration to book the most suitable tour for them. Therefore, by analyzing the game process between travel routes and consumers, we can analyze the influence of review features on consumers' purchase decisions and find an appropriate strategy to help consumers choose tourism products, and at the same time, help tourism companies improve the success rate of booking to increase revenue [16]. In analyzing consumers' purchasing decisions, consumers' utility can be used to represent their motivations for making choices. The more utility consumers have, the more likely they are to make purchasing decisions [17,18].

*2.2. Conceptual and Process Models of Consumer Decision-Making*

The characteristics of online reviews can have an impact on consumers' purchasing decisions [19,20]. For example, in the presence of fake reviews, Roman et al. [21] investigated the antecedents, consequences, and moderating factors of deceptive behaviors in online consumer reviews. Harrison-Walker and Jiang studied the impact of suspicious online reviews on reviewers' evaluations, consumers' attitudes toward brands and websites, and purchases, and found that consumers' suspicion that a review is fake will lead to a discount in reviewers' opinions, which in turn will negatively affect their attitudes toward brands and websites. Costa Filho et al. [22] believe that learning the four significant characteristics of fake reviews (one-sided, exaggerated, personal selling style, and general description) will affect consumers' trust in fake reviews and their perceived likability, thus affecting their purchase intention of target products.

Shah et al. [23] studied the influence of different factors of online peer review in an O2O food delivery application platform on consumers' persistent login behavior, investigated the role of text and pictograph online review content in inducing emotions, and believed that the PAD three dimensions of emotion were significant predictors of persistent login behavior. Sim et al. [24] used machine learning, natural language processing algorithms, and statistical methods to measure the impact of qualitative text reviews on accommodation booking intentions. In terms of manipulating online reviews, Zhuang et al.'s [25] analysis showed that the impact of adding positive reviews and deleting negative reviews on sales presented an inverted U-shaped curve [26,27]. Xu et al. [28] analyzed the influence of comment credibility and comment manipulation by comparing the comment data of some websites.

Due to the particularity of routes of online tourism products, spatial distance is a factor that most consumers need to consider before purchasing [29]. Therefore, spatial distance should be taken into account as an essential index when constructing the model. Among many other indicators of tourism products, price is generally an influential factor affecting consumers' purchasing decisions [30]. There have been many relevant studies, and price should also be considered an important indicator when considering modeling [31]. The characteristics of online reviews mainly include content characteristics, scores, number of

reviews, etc. These factors more or less have an impact on consumers' evaluations of the effectiveness of booking travel products [32]. However, in comparison, clear numbers, such as score and number of reviews, can form a stimulating effect more directly and quickly, thus affecting consumers' purchase decisions [33]. In addition, as content features are relatively comprehensive features, they are not easy to quantify and difficult to introduce in the modeling of consumer utility. Therefore, combined with the characteristics of online tourism products and the research focus of this paper, price, spatial distance, overall score, the number of reviews, and other indicators are taken as the main factors to consider when constructing the model [34].

The model is built to analyze the change in consumers' utility in the decision-making process of purchasing tourism products. The main factors considered in the modeling are all quantifiable [35]. Therefore, compared with the experimental method, the analysis method can obtain more objective results by using real data [36]. In the study of this paper, the real data are mainly used to obtain the price information of online tourism products through web crawler, but it is difficult to obtain consumer data [37]. Therefore, we model the consumption characteristics and game strategy of consumers, simulate a large number of consumers choosing whether to buy tourism products, and conduct big data simulation analysis [38]. The impact of online review features on consumer utility has been studied in the context of spatial distance and discount promotion of travel products [39].

## 3. Methodology

The present paper aimed to develop a model that examines the impact of hotel consumer purchase decisions. Specifically, the model will consider the influence of online comments (stimulus, S) and price adjustments on consumer decision-making. The behavioral response (R) will reflect the consumer's choice to either make a purchase or refrain from doing so, such as booking or not booking a travel product online. Additionally, the organic psychological response (O) will be explored as a comprehensive emotional and cognitive reaction during the purchase decision process. The construction of the model is depicted in Figure 1, outlining the fundamental concept [40].

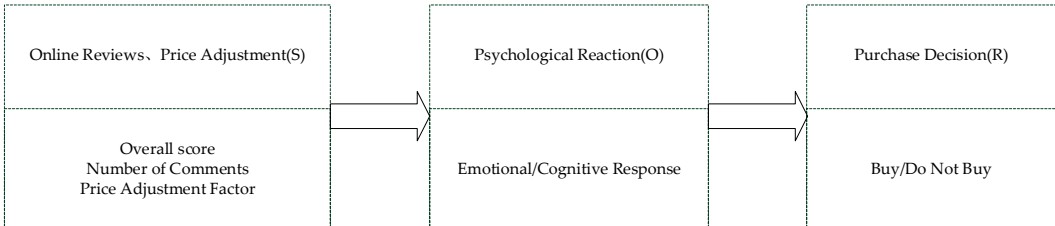

**Figure 1.** The idea of constructing the influence model of consumer purchasing decisions of tourism products.

After the initial screening, tourism products that meet consumers' basic needs will enter the alternative range (assuming there are N companies). At this stage, consumers will select the tour routes that are suitable for them from the N routes. For each potential consumer, there are N candidate tourism products, and each tourist route has three possible behavioral strategies [41]:

1. Accepting the existing price and making a direct booking. This strategy indicates that consumers fully accept the price of the tour route without hesitation.
2. Negotiating the price of the tourism products after booking. This strategy shows that consumers intend to book the tour line but consider the cost-effectiveness to be low. Therefore, they may negotiate the price. If the travel line reduces the price, consumers will accept the new price and proceed with the booking.
3. Not booking even after the price reduction of the travel route. This strategy indicates that consumers attempt to negotiate the price, but the new price does not meet their expectations. Therefore, they choose to book other travel route products.

The basic game model between consumers and hotels is shown in Figure 2.

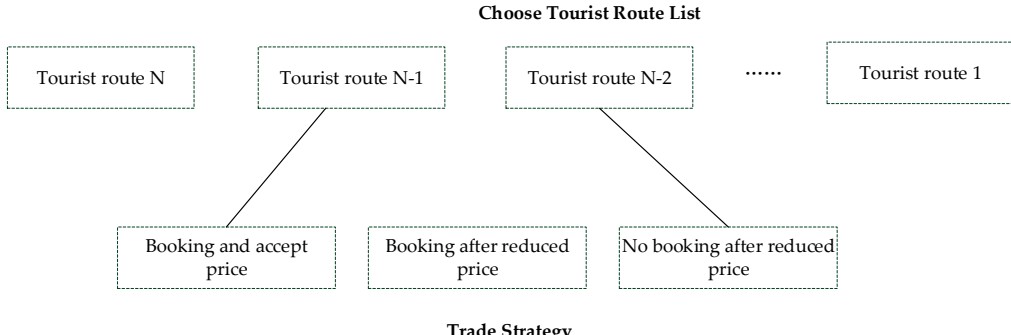

**Figure 2.** The basic game model between consumers and tourist routes.

The benefits derived from booking heavily influence consumer choices. As a result, consumers must assess and compare alternatives, necessitating the establishment of a consumer utility model to facilitate purchasing decisions.

Among the three consumer behavior strategies outlined above, price assumes a pivotal role in consumer decision-making processes. When making travel-related purchases, a tourist's final decision is contingent upon their perception of the price. However, it is crucial to note that price is not the sole determinant of consumers' decisions. Furthermore, various types of tourism products can influence consumers' purchase intentions through diverse means. In the context of booking travel routes, consumers offer comments and ratings on factors such as spatial distance, descriptive information, and performance indicators, all of which impact their decision-making. When selecting travel routes, consumers take all these indicators into account. Moreover, considering the features of online tourism products, an influential factor model was established, incorporating pricing, spatial distance, comprehensive ratings, review quantity, consumer strategy, and other indicators. This model establishes the relationship between the price function and the consumer utility function.

The analysis focused on the impact of travel route pricing on consumer decision-making. Price is a key factor influencing consumer choices, with varying levels of acceptability among different consumers. The price of each route ($M_i$) and the geographical orientation of tourists at the destination ($H_i = (x_i, y_i)$) are significant considerations. Spatial distance, representing the proximity of the consumer to the destination, is also a crucial factor. Additionally, a consumer's attribute score influences their route preference, which is calculated comprehensively by a third-party online platform. Time and cost are primary indicators, followed by mode of transportation and accommodation facilities. The final ranking of travel routes was determined by comparing attribute scores, with a lower value indicating a less favorable consumer evaluation. Consumer evaluations of various aspects of the route (place $E_{i1}$, facility $E_{i2}$, service $E_{i3}$, and hygiene $E_{i4}$) were averaged to obtain an overall evaluation ($E_i = (E_{i1} + E_{i2} + E_{i3} + E_{i4})/4$). A simple and practical model was adopted for the convenience of research in analyzing the number of reviews, which have both positive and negative effects on consumers. Let $N_i$ indicate the travel route. The number of reviews is normalized as: $L_i = \frac{N_i}{N_{max}}, L_i \in \text{N}[0, 1]$. When consumers book a trip, the game decision can involve choosing whether to wait for the price to fall, and the willingness of customers to reduce the price of travel routes is expressed using $\propto \in \text{N}[0,1)$.

Whether consumers choose one of the tourist routes depends on the utility when a consumer is choosing travel size, $U_s(\alpha)$, as indicated by Formula (1):

$$U_s(\alpha) = \delta\left(\alpha P_i - \frac{1+\alpha}{2}P_s\right), \tag{1}$$

The coefficient $\delta \in \text{N}[0,1)$ represents a utility equation and determines the scope of the entire equation. $P_i$ represents the standardized price of line I, while $P_s$ represents the

average standardized prices of other lines chosen by consumers. Formulas (2) and (3) correspond to $P_i$ and $P_s$, respectively:

$$P_i = L_i \log_2\left(1 + \frac{E_i M_i}{D_i}\right),\tag{2}$$

$$P_s = \frac{\sum\limits_{j \in I, i \neq 1} L_j \log_2\left(1 + \frac{E_i M_i}{D_i}\right)}{I - 1},\tag{3}$$

In Formulas (2) and (3), $E_i M_i$ are considered as contributing factors to the overall grading. The price of $E_i M_i / D_i$ reflects the standardization of space distance travel reservation price. $P_s$ represents the consideration of other potential factors, including the average price of the tourist route. $U_{bound}$ is used to determine whether consumers will book the travel, with $U_s(\alpha) > U_{bound}$ indicating that consumers will not hesitate to book the tour line. If $-U_{bound} \leq U_s(\alpha) \leq U_{bound}$, consumers will consider whether to reserve the tourist route. If $U_s(\alpha) < -U_{bound}$, it suggests that there are more suitable travel options, and the optimal choice for tourism consumers is to give up this line. The expression $(1 + \alpha)P/2$ approximates the opportunity cost for consumers to choose travel i, and if there is no negotiation with the travel, consumers will choose the tourism lines at price $M_i$.

## 4. The Game between Consumers and Travel Routes

If one delves deeply enough, consumers will ultimately select a specific route for booking, and the transaction process may encompass multiple stages, akin to numerous rounds of negotiations in a business setting. The aim of the game is to find the equilibrium between travel route providers and consumers, thus introducing a multi-stage game. This paper examines the two-stage game between tourist routes and consumers.

In these two stages, if the consumer agrees to the price of the travel route and makes a reservation in the first stage, then the second stage will not occur; otherwise, the second stage will unfold through a different $\delta$. The process of the two-stage game is illustrated in Figure 3.

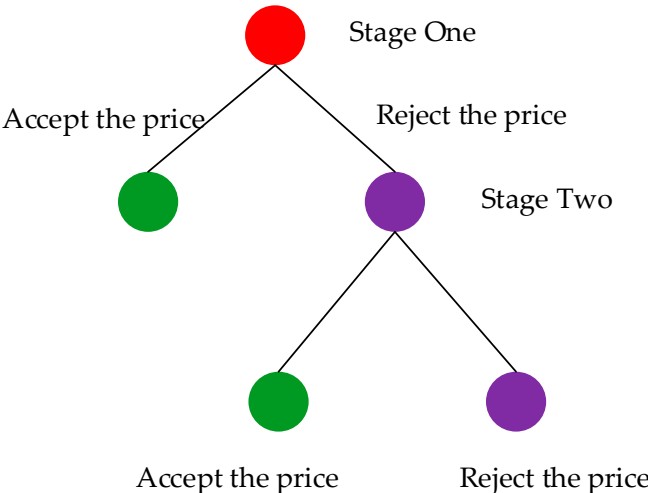

**Figure 3.** The two-stage game between the travel route and the consumer.

In the initial round of the game, consumers will opt to decrease $\delta_1$ to reduce $U_s(\alpha)$ while increasing their belief. When prices fall within the confidence interval, the consumer will proceed to the second round of the game, adjusting $\delta_1$ to $\delta_2$, thereby altering the utility equation, $U_s(\alpha)$. Figure 3 illustrates the sequential decisions and potential outcomes, where the consumer must decide whether to wait for a drop in trip prices. Despite the utility of

the consumer in the first round falling within the confidence interval, there remains the possibility that they may reject the price and enter the second round of the game [42].

It is assumed that prices and strategies undergo changes at different stages of the game, and that such changes are accessible, rendering the game efficient. In each round of the game, the consumer updates the expected price and utility function, eventually reaching equilibrium between the tour route and the consumer. The following presents each consumer's belief: $\mu_1(M) \in U[0, k]$ denotes the consumers' confidence in prices in the first round of games, while $\mu_2(M|a_1)$ represents the second game, where consumers reject $a_1$ following an update.

Assuming k is large enough, the upper limit of the price is defined as:

$$M^*(\alpha_1, \alpha_2) = \frac{2M_s(\alpha_1 - \delta_2\alpha_2)}{P_s[(1+\alpha_1) - \delta_2(1+\alpha_2)]}. \tag{4}$$

## 5. Game Strategy and Equilibrium

Since it is a two-stage game, in this section we outline the strategy for the initial round.

### 5.1. The Initial Stage of the Game Involving Consumers and Tourist Routes

**Lemma 1.** *Consumers will decline participation in the first-round game if any of the following conditions are satisfied:*

$$M \in \left[\frac{2\alpha_1 M_s}{P_s(1+\alpha_1)}, k\right], \text{ and } \alpha_1 > \alpha_2$$

$$M \in \left[M^*(\alpha_1, \alpha_2), k\right], \text{and } \delta_2\alpha_2 < \alpha_1 < \alpha_2$$

$$M \in [0, k], \text{ and } \alpha_1 < \delta_2\alpha_2$$

*Otherwise, the consumers will accept $\alpha_1$.*

**Proof.** In the two-stage game, consumers will choose different strategies to obtain varying utilities:

1. If customers accept $\alpha_1$, then:

$$U_s = \delta_1\alpha_1 M_s - \frac{1+\alpha_1}{2}\delta_1 MP_s$$

2. If consumers receive $\alpha_2$ and refuse $\alpha_1$, then the equation for the utility is:

$$U_s = \delta_1\delta_2\alpha_2 M_s - \frac{1+\alpha_2}{2}\delta_1\delta_2 MP_s$$

3. If customers refuse $\alpha_1$ and $\alpha_2$, then $U_s = 0$. Therefore, if:

$$\delta_1\alpha_1 M_s - \frac{1+\alpha_1}{2}\delta_1 MP_s > \delta_1\delta_2\alpha_2 M_s - \frac{1+\alpha_2}{2}\delta_1\delta_2 MP_s \tag{5}$$

consumers will choose to accept $\alpha_1$. Similarly, we can obtain:

$$\delta_1\alpha_1 M - \frac{1+\alpha_1}{2}\delta_1 MP_s > 0. \tag{6}$$

Obviously, consumers have the option to accept $\alpha_1$ instead of rejecting both $\alpha_1$ and $\alpha_2$ when $\alpha_1 > \delta_2\alpha_2$. Referring to Equation (5), there exists $0 < M < M^*(\alpha_1, \alpha_2)$. Taking into account Equation (6), the following conditions apply:

$$0 < M < min\left( M^*(\alpha_1, \alpha_2), \frac{2\alpha_1 M_s}{P_s(1 + \alpha_1)} \right). \tag{7}$$

When consumers choose to accept $\alpha_1$, they need to satisfy the following two conditions:
1. If $\alpha_1 > \alpha_2$, then:

$$0 < M < \frac{2\alpha_1 M_s}{P_s(1 + \alpha_1)}$$

2. If $\delta_2\alpha_2 < \alpha_1 < \alpha_2$, then:

$$0 < M < M^*(\alpha_1, \alpha_2)$$

Due to the fact that Equations (5) and (6) cannot be satisfied simultaneously, consumers will not accept $\alpha_1$ for any $M \in [0, k]$. Thus, the lemma is proven.

For the initial round of the game, any $\alpha_1$, the consumer's strategy choice is as follows: $\alpha_2^*(\alpha_1)$ are based on the fixed point of the equation for $\alpha_1$:

$$\alpha_2 = min\left( max\left( \alpha_p^*(k_1(\alpha_1, \alpha_2)), \frac{k_1(\alpha_1, \alpha_2)}{2M_s/P_s - k_1(\alpha_1, \alpha_2)} \right), min\left( \left| \frac{k}{\frac{2M_s}{P_s} - k} \right|, 1 \right) \right). \tag{8}$$

Among them,

$$\alpha_p^*(k_1(\alpha_1, \alpha_2)) = \sqrt{\frac{2M_s(2M - M_{bound})}{P_s M_r\left( \frac{2M_s}{P_s} - k_1(\alpha_1, \alpha_2) \right)}} - \frac{1}{2}, \tag{9}$$

$$k_1(\alpha_1, \alpha_2) = \frac{2M_s(\alpha_1 - \delta_2\alpha_2)}{P_s((1 + \alpha_1) - \delta_2(1 + \alpha_2))}. \tag{10}$$

The optimal solution for the initial round of the game is derived using the convex optimization toolbox:

$$\alpha_1^* = arg \max_{\alpha_1 \in [0,1]} \begin{pmatrix} (\delta_1(1 - \alpha_1)M_r - M_{bound})P_1 \\ +(\delta_1\delta_2(1 - \alpha_2^*(\alpha_1)M_r - M_{bound}))P_2 \\ +(\delta_1\delta_2 - 1)M_{bound}P_3 \end{pmatrix}. \tag{11}$$

The optimal solution formula for the probability of each policy, denoted as $P_1, P_2$, and $P_3$, is determined when $\delta_2\alpha_2^*(\alpha_1^*) < \alpha_1^* < \alpha_2^*$. At this point, belief and strategy constitute the initial round of the equilibrium solution:

$$P_1 = \frac{k_1(\alpha_1)}{k}$$

$$P_2 = \frac{\left( \frac{2M_s\alpha_2}{P_{s(1+\alpha_2)}} - k_1(\alpha_1) \right)}{k}, P_3 = \frac{\left( k - \frac{2M_s\alpha_2}{P_{s(1+\alpha_2)}} \right)}{k} \tag{12}$$

$\square$

## 5.2. The Second Stage of the Game between Consumers and Tourist Routes

Similar to the first game, we continued with the analysis of the second game.

**Lemma 2.** *The following beliefs and strategies form an infinite set of equilibrium solutions:*

$$\alpha_2^* = min\left( max\left( \alpha_p^*, 0\right), min\left( \left| \frac{k}{\frac{2M_s}{P_s} - k} \right|, 1\right)\right),\tag{13}$$

$$\alpha_p^* = \sqrt{\frac{2M_r - M_{bound}}{M_r}} - \frac{1}{2}.\tag{14}$$

*Each parameter must meet the following conditions:*
*1. $\alpha_1^*$: For any positive number, $\alpha_1^* < \delta_2 \alpha_2^*$*
*2. $\mu_2(M) = \mu_2(M|\alpha_1)$: Consumer $\mu_1(M) \in U(0,k)$ obeys normal distribution*
*3. $\aleph_1(\alpha_1|M)$: Consumer refused $\alpha_1$*
*4. $\aleph_2(\alpha_2|M, \alpha_1)$: If $\delta_1\delta_2\alpha_2 M_s - \frac{1+\alpha_2}{2}\delta_1\delta_2 P_s M > 0$, consumer accepted $\alpha_2$*

**Proof.** If $M \in [0,k]$ and $\alpha_1 < \delta_2\alpha_2$ in the initial round of the game, the value of the second game is significantly higher. Hence, consumers will reject $\alpha_1$. Consequently, the optimal strategy for consumers in the subsequent round of the game is as follows:

$$\alpha_2^* = min\left( max\left( \alpha_p^*, 0\right), min\left( \left| \frac{k}{\frac{2M_s}{P_s} - k} \right|, 1\right)\right),\tag{15}$$

$$\alpha_p^* = \sqrt{\frac{2M_r - M_{bound}}{M_r}} - \frac{1}{2}.\tag{16}$$

Thus, $\alpha_2^*$ is independent of $\alpha_1$, and when the other parameters are constant, $\alpha_2^*$ remains constant. Thus, when a given $\alpha_1$ is provided, the consumer strategy is as follows:

*1. $\alpha_2^* = min\left( max\left( \alpha_p^*, 0\right), min\left( \left| \frac{k}{\frac{2M_s}{P_s} - k} \right|, 1\right)\right)$ is constant*
*2. $\mu_1(M) = \mu_2(M|\alpha_1)$:$\mu_1(M), \mu_2(M) \in U(0,k)$ obeys normal distribution*
*3. $\aleph_1(\alpha_1|M)$: Consumer refused $\alpha_1$*
*4. $\aleph_2(\alpha_2|M, \alpha_1)$: If $\delta_1\delta_2\alpha_2 M_s - \frac{1+\alpha_2}{2}\delta_1\delta_2 P_s M > 0$, the consumer accepted $\alpha_2$*

In accordance with previously stated formula, variables $P_1, P_2, P_3$ in Equation (12) are transformed as follows:

$$P_1 = 0$$

$$P_2 = \frac{2M_s\alpha_2^*}{kP_s(1+\alpha_2^*)} P_3 = 1 - \frac{2M_s\alpha_2^*}{kP_s(1+\alpha_2)},\tag{17}$$

The optimal $\alpha_1 \in N[0,1]$ will be solved via the efficiency maximization solution:

$$U_p(\alpha_1) = (\delta_1\delta_2(1 - \alpha_2^*)M_r - M_{bound})\frac{2\alpha_2^* M_s}{kP_s(1+\alpha_2^*)} + (\delta_1\delta_2 - 1)M_{bound}\left(1 - \frac{2\alpha_2^* M_s}{kP_s(1+\alpha_2^*)}\right).\tag{18}$$

Given that $\alpha_1$ is absent from Equation (18), let $\alpha_1^*$ be any number in the interval $N[0,1]$ that is greater than zero, as required for the lemma's proof. $\square$

## 6. Simulation and Result Analysis

In this section, we performed data simulation on the interaction between consumers and hotels, utilizing big data analysis to incorporate a substantial amount of historical hotel data. This was carried out to analyze the impact of the previously mentioned optimal game strategy.

During the model construction, we established the comprehensive parameter ratings, $E_i \in [1, 5]$, and the standardized comment count, $L_i \in N[0, 1]$. In the simulation phase, we will demonstrate the influence of historical data through these two parameters. The historical hotel price data were acquired through web crawling. The data structure design is illustrated in Figure 4.

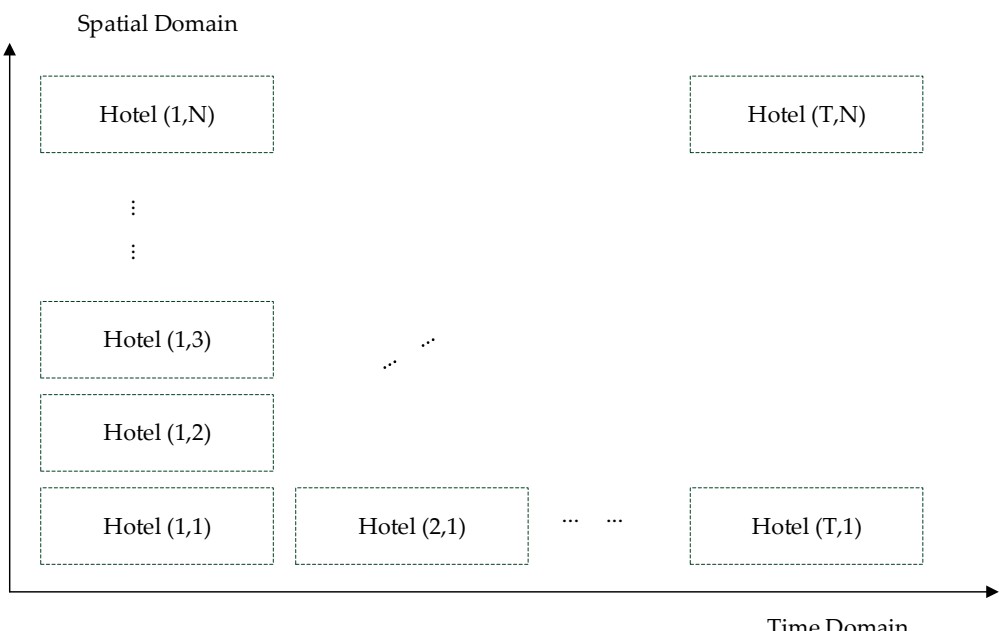

**Figure 4.** Hotel price historical data structure.

In Figure 4, the two-dimensional dataset comprises time and hotel information. Specifically, it includes the daily arrivals for each hotel.

Before commencing the simulation, it is essential to establish the simulation parameters. The variable M is defined as $K = 1.5 \, M_s / P_S$, representing the ceiling value. $\delta_1 = 0.5$, $\delta_2 = 0.7$, indicating the relaxed game conditions for consumers and tourists in the second stage. The interests of consumers are expanded through the setting $M_{bound} = 1$, $M_s = 10$, $M_r = 200$. For the convenience of the study, the number of consumers was set at 1000, and data from 200 travel routes between 24 May 2021 and 23 May 2022 on Ctrip were accurately captured. (see Appendix A) To simplify the study and mitigate the impact of low-quality tour itineraries, all selected hotels were required to have an overall rating of at least four stars.

Two simulation experiments were conducted. The primary objective of the first experiment was to investigate the correlation between the overall score and the change in average consumer benefits resulting from a reduction in the price of tourist routes. The second experiment aimed to explore the relationship between the number of reviews and the change in average consumer benefits following a reduction in the price of a travel route.

For the first simulation experiment, four groups were established. The first group involved the random generation of all data. In the second group, $L_i$ was fixed, where $L_i = \frac{\sum_{i \in I} L_i}{I}$, and the remaining data were randomly generated. The third set had a fixed $L_i$, and the ratings of 200 tourist routes were computed based on the scores of the 100 tourist routes with the lowest actual parameter values. The fourth group maintained a fixed $L_i$ and calculated the high scores of 100 tourist routes based on the actual parameter values.

The simulation results are shown in Figure 5.

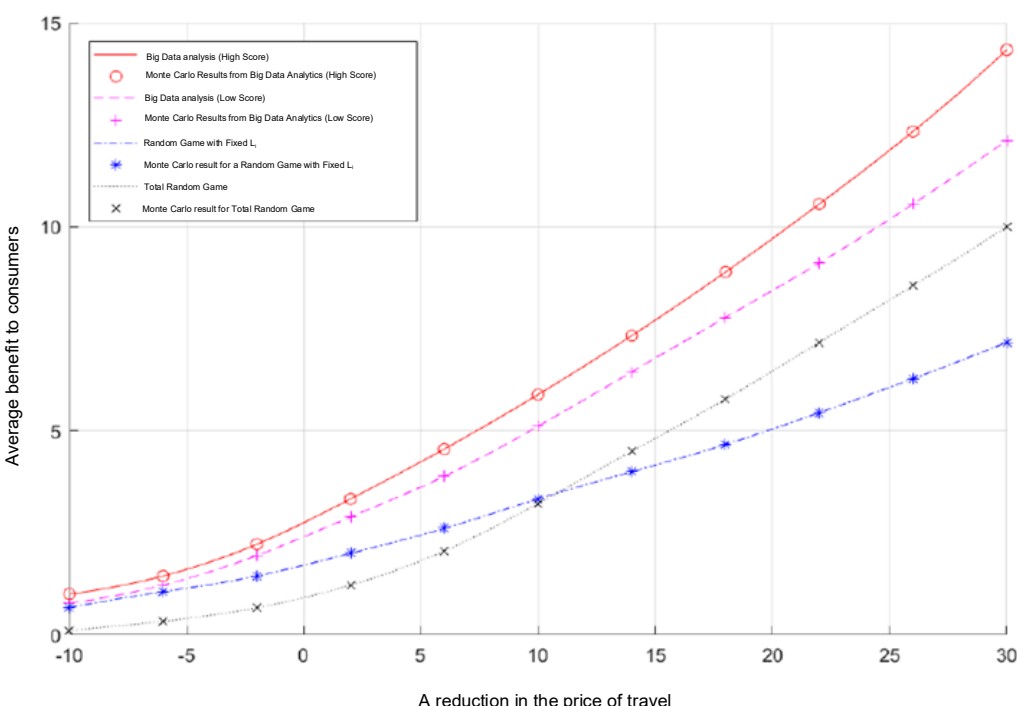

**Figure 5.** The results of the first simulation experiment.

In Figure 5, the *X*-axis represents the standardized reduction of the price of tourist routes, and the *Y*-axis represents the average income obtained by consumers through the game, expressed by a utility function. The figure presents four sets of results, each including theoretical results and Monte Carlo results. The red curve of big data analytics represents the overall score in the higher case, while the pink dotted line represents the simulation results. The use of big data analysis provided the overall rating of the simulation results in the lower case, while the blue dashed line represents the fixed $L_i$ random game under the simulation results. The black dotted line represents all the simulation results under the random game.

In the second simulation experiment, four groups were also analyzed, and all the data in the first group were randomly generated. The second group had a fixed $E_i$, where $E_i = 3.5$, and the data were randomly generated. The third group had a fixed $E_i$ and involved the ordering of 200 tourist lines by the number of comments, taking the lower 100 tourist routes' actual parameter values into account. The fourth group had a fixed $E_i$ and involved the selection of 100 highly commented on tourist routes' actual parameter values for calculation. The simulation results are presented in Figure 6.

In Figure 6, the *X*-axis depicts the standardized reduction in the price of tourist routes, while the *Y*-axis represents the average income obtained by consumers through the game, expressed by the utility function. Similar to the initial simulation, the results were divided into four groups, each comprising theoretical results and Monte Carlo results. The red curve represents the simulation results in the scenario of a substantial number of comments analyzed with big data. The pink dotted lines depict the simulation results for a low number of comments obtained through big data analysis. The blue dashed line represents the simulation results for the fixed $E_i$ random game. The black dotted line represents the simulation results across all stochastic games. These results are consistent with those depicted in Figures 5 and 6.

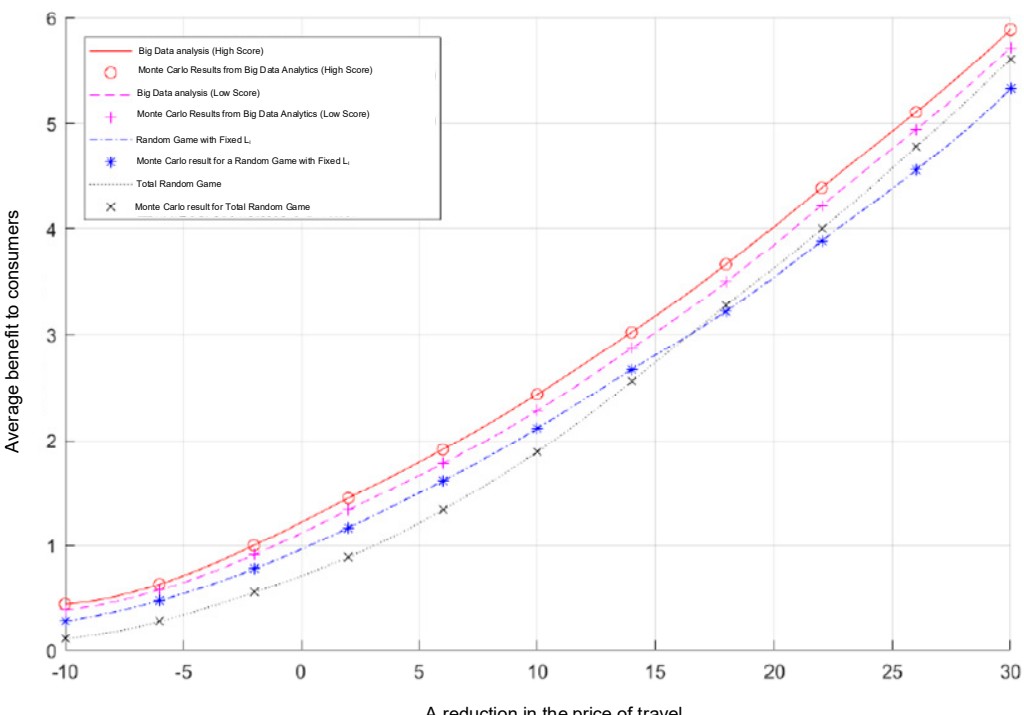

**Figure 6.** The results of the second simulation experiment.

The Monte Carlo simulation data aligned with the data from the theoretical analysis.

The model proposed in this paper exhibited superior performance due to the two-stage game, which provides consumers with an additional opportunity to choose and the tour route with an extra chance to attract customers. Consequently, the second round of the game enhanced the likelihood of consumer bookings. The approach advocated in this paper is anticipated to yield greater advantages in the game's second stage.

In the absence of an analysis based on historical data, consumers are unable to dynamically adjust their strategies according to the historical data of travel routes, resulting in a two-stage game process. Figures 5 and 6 demonstrate that this method produced similar effects to the random method when the price was reduced by 10 and 18 points, respectively. However, as the price continued to decrease, the efficacy of this method diminished in comparison to the random method. This may be attributed to the non-cooperative nature of the game between tourist routes and consumers. Through the two-stage game, consumers remained uncertain about the acceptability of the price, leading to an increased likelihood of consumers declining reasonable prices for tourist routes, thereby diminishing the effectiveness of this method beyond the inflection point.

As depicted in Figure 5, when the price of a travel route decreased, the overall score exerted a positive influence on the average benefit of consumers. A comparison between the curves of high and low scores revealed that the game effect between consumers and tourist routes was more favorable when the overall score of tourist routes was high, indicating a significant impact of the overall score of tourist routes on consumers' utility.

As depicted in Figure 6, an increase in the number of reviews positively influenced the average consumer benefit when the price of travel routes decreased. A comparison of the two curves for high and low numbers of comments revealed that with a high number of comments, the interaction between consumers and tourist routes was marginally better than when the number of comments was low, although the difference was not significant.

This phenomenon can be attributed to several factors. Firstly, the number of comments not only reflects the popularity of the tour route but also leads consumers to anticipate large crowds or inadequate service at the tourism destination due to the high volume of tour groups. Secondly, an abundance of comments also increases the likelihood of negative feedback, which offsets some of the advantages brought by a high number of comments.

Thirdly, excessive comments may result in information overload, causing some consumers to opt out.

In conclusion, both the overall score and the number of comments positively impacted consumer decision-making during the purchase stage. However, the overall score had a greater influence on consumer decision-making than the number of comments. As the number of comments on travel routes accumulates, routes should select online comments that provide consumers with effective and accurate information to display among the multitude of comments. Additionally, by enhancing service, environment, sanitation, facilities, and other conditions, tourist routes can guide and encourage consumers to give higher scores to positively influence consumer behavior and improve the booking rate of tourist routes.

## 7. Conclusions

In this paper, we presented a fundamental non-cooperative game model for consumers' purchasing decisions regarding tourist routes. The utility level of consumers during the booking process was utilized to represent their purchasing decisions. A two-stage game model between consumers and tourist routes was formulated and examined, and corresponding game strategies were suggested. Utilizing the acquired tourism big data, a simulation experiment was conducted to analyze the impact of overall ratings and the number of reviews on consumers' utility when the price of the tour route changed. The simulation results demonstrated that both the overall rating and the number of reviews had a positive effect on consumer utility, with the overall rating exerting a greater impact than the number of reviews. Finally, here, the above conclusions are briefly analyzed, and several strategies are proposed regarding the presentation of online reviews and guiding consumers to score, which can help enhance the efficiency of online booking platforms.

We aimed to investigate the influence of online reviews on consumers' decision-making process when purchasing travel routes. Specifically, we examined the impact of the overall rating and number of reviews on consumers' preferences in the context of discounted travel route prices. Since the overall score is derived from multiple attribute scores, it may be more relevant to consider each attribute score as a parameter in the consumer utility function. Furthermore, in addition to the quantity of reviews, the content of the reviews may also play a role in influencing consumer preferences. As a result, future research will seek to enhance the model by integrating review content with attribute scores to yield more precise calculations. This will allow for a deeper exploration of the impact of online reviews on consumer purchase decisions and an analysis of various hotel consumer types based on factors such as review quantity and content. The study will also examine their influence on post-purchase recommendation decisions.

This paper was centered on the central issue of how online reviews influence hotel consumers' decision-making processes, particularly in the context of their purchase decisions. By integrating the online experience of hotel products and the geographical characteristics of their locations, a model was developed to depict the impact of hotel and consumer interactions on purchase decisions. This model aims to assess the utility of hotel consumers in making reservations for hotel products and to analyze how online reviews influence consumer purchase decisions amid fluctuations in hotel prices.

This study was confined by limitations in research conditions, time, manpower, and data acquisition. As a result, we focused solely on the impact of certain key factors, and there were constraints in addressing certain issues. In future research, the scope can be expanded to encompass the following aspects.

In light of the influence of online reviews on hotel consumers' purchasing decisions, we focused on the effects of overall hotel ratings and the number of reviews on consumer utility when hotel prices are reduced. However, the overall rating is derived from the scores of multiple hotel attributes, suggesting that it might be more beneficial to consider the scores of various hotel attributes as parameters of the consumer utility function. Furthermore, in addition to the number of reviews and ratings, the content of reviews may also impact

consumers' utility. Therefore, future improvements to the model could involve integrating the text of review content and ratings of various hotel attributes to obtain more precise calculation results and delve deeper into the impact of online reviews on consumers' purchasing decisions.

We examined the user recommendation rate in conjunction with the differences in hotel properties. The hotel property differences mentioned here pertain to consumers' varying ratings for each property based on their individual experiences after check-in. Alternatively, consumers may also form their initial inclination toward a hotel based on the cumulative scores of various attributes on the online hotel booking platform, representing the purchase intention in previous studies. Consequently, subsequent analysis can be conducted on the influence of hotel property differences on the purchase intention of different consumer segments.

**Funding:** This research was funded by Natural Science Foundation of China (71531005) and the Program of the Shanghai Academic Research Leader (20XD1400400).

**Institutional Review Board Statement:** Ethical review and approval were waived for this study due to the anonymous and voluntary participation in this survey.

**Informed Consent Statement:** Informed consent was obtained from all subjects involved in the study.

**Data Availability Statement:** The datasets used and/or analyzed during the current study are available from the corresponding author upon reasonable request.

**Acknowledgments:** We thank the anonymous reviewers and the editor who helped improve the scientific quality of the original manuscript.

**Conflicts of Interest:** The authors declare no conflicts of interest.

## Appendix A

**Table A1.** Particulars of Ctrip's hotel data.

| Date | Hotel Name | Price | Discounted Price | Pre-Determined Number | Rate of Recommendation | Number of Comments | Ctrip Customer Ratings | Star Rating | Average Guest Rating |
|------|-----------|-------|------------------|----------------------|----------------------|-------------------|----------------------|-------------|---------------------|
| 18 February | Yan'an Hotel | 896 | 447 | 20 | 97% | 2619 | 4.6 | 4 | 4.7 |
| 18 February | Crystal Orange Shanghai Hongqiao Hub International Exhibition Center Hotel | 440 | 350 | 15 | 98% | 2590 | 4.9 | 4 | 4.9 |
| 18 February | Crystal Orange Shanghai Jiangqiao Wanda Hotel | 469 | 398 | 18 | 96% | 731 | 4.8 | 4 | 4.7 |
| 18 February | Mehow Elegant Shanghai Jiading New Town Center Hotel | 605 | 399 | 14 | 96% | 1823 | 4.6 | 4 | 4.7 |
| 18 February | Bvlgari Hotel Shanghai | 5364 | 5314 | 26 | 97% | 1193 | 4.7 | 5 | 4.7 |
| 18 February | Shanghai Hongqiao Hotel | 1120 | 448 | 6 | 96% | 3596 | 4.6 | 4 | 4.7 |
| 18 February | Yilin Junting Hotel, Caohéjīng, Xuhui, Shanghai | 711 | 497 | 19 | 95% | 1175 | 4.7 | 4 | 4.7 |
| 18 February | Shanghai Hongqiao National Exhibition and Convention Center Tonp Hotel | 497 | 347 | 5 | 95% | 401 | 4.8 | 4 | 4.8 |
| 18 February | Shanghai Jing'an Shangri-La Hotel | 2085 | 2017 | 39 | 97% | 2920 | 4.7 | 5 | 4.7 |
| 18 February | Shanghai Bund Old Wharf CitiGO HuanGe Hotel | 538 | 505 | 17 | 96% | 1551 | 4.7 | 4 | 4.6 |
| 18 February | basePLUS Serviced Apartments (Shanghai Binjiang Branch) | 783 | 500 | 9 | 97% | 401 | 4.6 | 4 | 4.6 |
| 18 February | Ramada Encore by Wyndham Shanghai Pudong Airport | 528 | 395 | 21 | 99% | 456 | 4.7 | 4 | 4.7 |
| 18 February | Meichengli Hotel (Shanghai Hongqiao Hub National Exhibition Center) | 797 | 507 | 17 | 96% | 3648 | 4.8 | 4 | 4.8 |
| 18 February | City Home Apartment (Shanghai Lujiazui Expo Park Branch) | 748 | 513 | 6 | 96% | 2173 | 4.7 | 4 | 4.7 |
| 18 February | Shanghai Zhongtie Wanxin Hotel | 583 | 415 | 3 | 97% | 1216 | 4.9 | 4 | 4.9 |

**Table A1.** *Cont.*

| Date | Hotel Name | Price | Discounted Price | Pre-Determined Number | Rate of Recommendation | Number of Comments | Ctrip Customer Ratings | Star Rating | Average Guest Rating |
|---|---|---|---|---|---|---|---|---|---|
| 18 February | Kanghong Garden Hotel | 640 | 416 | 8 | 96% | 211 | 4.5 | 4 | 4.6 |
| 18 February | Shanghai Shaanxi Business Hotel | 541 | 459 | 19 | 97% | 1166 | 4.4 | 4 | 4.5 |
| 18 February | Jinjiang Metropolo Hotel (Shanghai Wujiaochang) | 504 | 428 | 24 | 95% | 842 | 4.8 | 4 | 4.8 |
| 18 February | Shanghai Happy Family Hotel | 625 | 437 | 18 | 96% | 894 | 4.5 | 4 | 4.5 |
| 18 February | The Bund W Shanghai | 2658 | 2598 | 32 | 98% | 3028 | 4.5 | 5 | 4.5 |
| 18 February | Yilin Junting Hotel in Caohéjīng, Xuhui, Shanghai | 711 | 497 | 13 | 97% | 1175 | 4.7 | 4 | 4.8 |
| 18 February | Yi Fei Hotel, Jin Qiao Center, Shanghai | 909 | 453 | 19 | 96% | 185 | 4.8 | 4 | 4.8 |

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
