# Peer review of "The Influence of Online Reviews on the Purchasing Decisions of Travel Consumers"

_sustainability, doi:10.3390/su16083213_

Round 1

Reviewer 1 Report

Comments and Suggestions for Authors

While I find the topic of the paper to be both interesting and relevant, the manuscript as a whole encounters several issues that need addressing.

The introduction and literature review are in the same chapter (Introduction). I suggest separating these two sections in the manuscript and providing a more detailed description of the current state of the issue in the introduction.

I believe the author struggles with English throughout the entire paper, which somewhat diminishes the article's explanatory value. Some chapter’s headings begin with lowercase letters, while others begin with uppercase letters, and this alternation occurs inconsistently. Some sentences throughout the text are not correct in English, or the author uses incorrect expressions. For example, the sentence in line 117 states: "Combined with the characteristics of online tourism products and the research content of this paper, this paper will select the price, spatial distance, overall score, number of reviews, and consumer strategy of tourist routes and other indicators to model the influencing factors and to model the price function and consumer utility function." Particularly, the expression "this paper will select the price..." is ambiguous. Did the author intend to convey that "the author will select the price," since it seems unlikely that a paper itself could select the price?

Or another sentence:  Consistent with the spatial distance of the first stage of consumer decision refers to the distance between the hotel and the destination home input by the consumer.“  (lines 125-127. Please correct the expressions throughout the entire manuscript because there are instances where the meaning of the sentence becomes a little bit confusing.

In lines 121, 125, 130, 137, and 140, I recommend writing complete sentences instead of fragmented phrases such as "First, the price of the tour route. Second, space distance," etc.

Many paragraphs throughtout the text are not very well written and than the whole meaning of these sentences is unclear (for example lines 138-156, 235-240, 260-265, 266-271.....)

In the section Materials and Methods, there is a lack of description or rationale for the chosen methodology. Furthermore, the author does not acquaint us with the selection of the method he decided to use in his scientific work. Author directly describe input indicators.

Only in the section Simulation and Result Analysis do I learn that it is a "set of results including theoretical results and Monte Carlo results." The results of the work are general and lack high explanatory value. I suggest the author clearly describe the conclusions reached once again. The objective of this chapter is not clear; it seems somewhat detached from the context. The author in this section limits themselves to describing the results of the program used, but the work requires deeper discussion and debate.

The references is not in the correct order throughout the entire text of the manuscript and the references form is not consistent with the journal's requirements.

In Conclusion line 348 there is a duplicaly written „in additon“.

Comments on the Quality of English Language

I believe the author struggles with English throughout the entire paper, which somewhat diminishes the article's explanatory value.

Author Response

Thank you fro your letter and reviewer's comments concerning our manuscript entitled "Impact of Online Reviews on Travel Consumer’s Purchasing Decisions".    Those comments are valuable and very helpful.    We have read through comments carefully and have made corrections.    Based on the inscruction provided in your letter, we uploaded the file and the revised manuscript.

I have provided in-depth responses to the queries raised in the attached document.

We would love to thank you allowing us to resubmit a revised copy of the manuscript and we highly appreciate your time and consideration.

Reviewer 2 Report

Comments and Suggestions for Authors

The manuscript examine the influence of online review characteristics on consumers' purchasing decisions within the framework of spatial distance. It is obvious that the research topic is interesting. Overall, the research approach of the manuscript is clear, the logical reasoning is rigorous, and the research conclusions are correct. I think the following modifications are needed before accepting this manuscript.

1. It is suggested that the author further clarify why he chose this research method to study this problem, and how applicable and progressiveness this method is.

2. It is recommended to further supplement the literature review section with research results of the same type in the past three years. Currently, some references are relatively outdated.

Author Response

(The authors gave the same response as above.)

Reviewer 3 Report

Comments and Suggestions for Authors

1. In this article, the literature review is in the introduction. In our opinion, we can separate them or add in the introduction a part about the relevance of this study. You also need to clearly state the goals and usefulness of this work.

2. We can recommend that the authors add discussion questions in the results section.

Author Response

(The authors gave the same response as above.)

Round 2

Reviewer 1 Report

Comments and Suggestions for Authors

The author has incorporated all of the reviewer's comments and suggestions into the revised manuscript. The English language has been thoroughly reviewed and improved. Additionally, the Introduction, Methodology, Results, and Conclusion sections have been expanded and clarified. A new chapter, Literature Review, has been added, addressing a previously missing component.